# Recovery and Preparation of Potassium Fluorotantalate from High-Tantalum-Bearing Waste Slag by Pressure Alkaline Decomposition

**Kangde Xie [1], Xiuyu Wei [1,2], Longgang Ye [1,2,3,*], Mingyuan Wan [1,2], Shilin Li [1] and Jianguo Wu [1]**

1    Zhuzhou Cemented Carbide Group Co., Ltd., Zhuzhou 412000, China; xiekd@minmetals.com (K.X.); weixiuyu79@126.com (X.W.); 15388017911@163.com (M.W.); lisl@601.cn (S.L.); www6219@163.com (J.W.)
2    State Key Laboratory of Cemented Carbide, Zhuzhou 412000, China
3    College of Materials and Advanced Manufacturing, Hunan University of Technology, Zhuzhou 412007, China
*    Correspondence: yelonggang@sina.cn; Tel./Fax: +86-0731-22183453

**Abstract:** Tantalum slag is a type of high-grade tantalum resource with great recovery value. In this paper, a low fluorine process, including alkali pressure decomposition, low-acid transformation, solvent extraction, and crystallization, is proposed to recover tantalum and prepare potassium fluotantalate. First, some tantalum slag underwent alkali pressure decomposition, and the optimal decomposition conditions were obtained under a reaction time of 2 h, oxygen partial pressure 2.5 MPa, liquid–solid ratio 4:1, basicity 40 wt.%, and temperature 200 °C. Under these conditions, the decomposition efficiencies of Ta and Nb were 93.62% and 95.42%, respectively. X-Ray diffraction (XRD) and scanning electron microscope (SEM) were used to detect the main phase of the decomposition residue and showed that it was mainly sodium tantalate. With the increase in oxygen partial pressure, the particle size of decomposition slag gradually decreases and becomes loose. Second, the alkali decomposition residue was subjected to low-acid leaching to obtain fluorine tantalate and fluorine niobate, and the leaching efficiencies of tantalum and niobium were more than 99%. Last, the low-acid leaching solution was subjected to solvent extraction and evaporative crystallization to prepare potassium fluotantalate. The results showed that the tantalum extraction rate and tantalum and niobium separation factors were above 94% and 200, respectively, and the purity of potassium fluotantalate met the requirements of commercial products. Compared with current industrial practice, the consumption of hydrofluoric acid was greatly reduced, and the recovery rate of tantalum was increased.

**Keywords:** wastematerials of tantalum; pressure leaching; alkali decomposition; sodium tantalate; recycling

## 1. Introduction

Tantalum and niobium are important rare refractory metals with high melting points, corrosion resistance, and good machining performance [1,2].They are widely used in electronics, metallurgy, iron and steel, the chemical industry, carbide, and other technical fields [3]. The properties of niobium and tantalum and their compounds are very stable, generally insoluble in inorganic acids except for HF acid, but easily react with alkali to form a series of basic salts. In the application field of tantalum and niobium, potassium fluotantalate and niobium hydroxide are the main intermediate products, which are extracted from tantalum and niobium materials and can be reduced and processed into various tantalum and niobium products [4,5].

Raw materials for tantalum and niobium extraction include primary minerals, such as tantalite, coltan, pyroclasts, and niobite, and secondary resources, such as decomposition residue, tungsten residue, tin residue, and processing waste [6–9]. With the decrease in primary resources, more attention is given to the recovery of secondary tantalum and

niobium resources. At present, the main extraction methods of tantalum and niobium can be divided into acid decomposition and alkali decomposition. Acid decomposition media include nitric acid [10,11], hydrochloric acid [12,13], sulfuric acid [14], hydrofluoric acid [15], and sulfuric acid-hydrofluoric acid mixed acid [16,17]. The acid leaching solution is subjected to solvent extraction to separate and purify tantalum and niobium.

Hydrofluoric acid is the most commonly used leaching agent. Concentrations of 40–48% hydrofluoric acid can leach 90% of Ta and Nb from raw ore, forming complexes $TaF_7^{2-}/TaF_6^-$ and $NbOF_5^{2-}/NbF_6^-$. The leaching reaction is usually conducted at high temperatures (80–100 °C) and acid concentrations (up to 20% hydrofluoric acid, HF) [18]. However, hydrofluoric acid leaching is only suitable for high-grade tantalum and niobium ores, which have high acid consumption for low-grade ores. Meanwhile, hydrofluoric acid is easily volatilized, resulting in losses and environmental hazards. It is also highly corrosive to industrial equipment and difficult to treat in fluorinated wastewater. Therefore, researchers proposed introducing sulfuric acid to reduce the consumption of hydrofluoric acid. Zhu and Cheng [19,20] pointed out that the $HF-H_2SO_4$ mixture also promoted Ta/Nb leaching better than HF leaching alone, and the operating temperature of $HF-H_2SO_4$ leaching was usually kept at approximately 100 °C. Yang Jihong and Yang Xiuli [21,22] investigated the decomposition of tantalum and niobium ore by pressure leaching by $HF-H_2SO_4$ mixed acid, which could effectively improve the decomposition rate of tantalum and niobium and reduce the amount of HF, thus improving the operating environment. Zhao Mingzhi [23] treated titanium–tantalum–niobium ore with $HF-H_2SO_4$ acid and improved the tantalum–niobium decomposition rate to more than 97% by reducing the particle size of the ore powder, increasing the amount of the acid, and increasing the decomposition time. To eliminate the consumption of hydrofluoric acid, Kabangu [24] proposed decomposing tantalum ore by molten ammonium difluoride at 200–400°C, leaching with water to obtain $(NH_4)_2TaF_7$ and $(NH_4)_3NbOF_6$, and then using solvent extraction to recover Ta and Nb.

There are two kinds of alkali decomposition methods: molten alkali decomposition and sub-molten salt decomposition [25]. Due to the easy alkali reaction of tantalum and niobium, various compounds, such as caustic soda, potassium hydroxide, potassium carbonate, or their mixtures, are proposed for the alkali decomposition of tantalum and niobium. Zheng Shili [26,27] used KOH to decompose low-grade Ta and Nb ores in a high-concentration KOH solution and then recovered Ta and Nb by water leaching and solid–liquid separation. This method can treat low-grade tantalum and niobium ores without HF. Wang [28] further recovered tantalum and niobium from tantalum and niobium waste residue by the all-wet process of low alkali decomposition—water leaching, dilute acid pretreatment, and hydrofluoric acid transformation leaching. Under optimized conditions, the recovery rates of tantalum and Nb reached 98.37% and 99.15%, respectively. Nguyen and Lee [29] also performed tantalum and niobium extraction from low-grade tantalum and niobium ores by alkali decomposition and low-acid leaching. Adekola [30] proposed the leaching of tantalum and niobium from natural ores by combining sodium nitrate and sodium peroxide. Wang [31] once fused $Nb_2O_5$ with potassium oxide at 400 °C and obtained $K_8(Ta,Nb)_6O_{19}$ salt by water leaching, evaporation, and crystallization, which was then converted into high purity niobium oxide and tantalum oxide $(Nb,Ta)_2O_5$ by dilute acid. The alkali decomposition process has a high decomposition rate and reduces the consumption of hydrofluoric acid.

In the tantalum and niobium profiles and purity process in the bombarding furnace, due to bombardment of electron beam and high-temperature volatilization, a certain amount of metals are deposited on the inner wall of the furnace, so the hanging fireplace slag is produced. Due to the variety of processed materials, the composition and phase of slag are complicated, but the tantalum content is high and has a great recycling value. Based on the advantages of low fluorine and high transformation efficiency of the alkali decomposition method, this paper proposes decomposing this tantalum waste by pressurizing alkali to obtain easily decomposed sodium tantalate and sodium niobate, converting tantalum and niobate into soluble tantalum and niobate by low-acid leaching, and then recovering

tantalum and producing potassium fluotantalate by extraction and crystallization. The decomposed lixivium containsunreactedexcess alkali and can be returned to decomposition for recycling. Compared with the melt alkali decomposition process, it can cancel the alkali enrichment process from decomposed lixivium and reduce alkali consumption, with clean and low consumption characteristics. In this paper, the effects of various operating conditions on the decomposition of tantalum and niobium are studied in detail.

## 2. Experimental

### 2.1. Materials and Instruments

The high-tantalum-bearing waste slag used in the experiment was generated during the processing of tantalum and niobium section materials in an electro-bombardment furnace. Its main components are shown in Table 1. The tantalum content was up to 90.85%, the niobium content was also up to 5.89%, and the main impurity elements were Ca, Fe, Na, Al, and Ca. The slag was pulverized by hydrogenation to obtain a fine powder, and the XRD and laser particle size distribution of the Ta residue is shown in Figure 1. The main phases of tantalum were$Ta_2O_5$, TaC, and $FeTaO_3$, with a small amount of Nb and Si carbide, and the particle size of 90% of the powder slag was less than 350 μm.

**Table 1.** The content of each element in tantalum slag.

| Element | Ta | Nb | Ca | Ti | Si | Fe | Cu | Al |
|---|---|---|---|---|---|---|---|---|
| Content/% | 72.41 | 3.42 | 2.61 | 1.88 | 1.44 | 1.27 | 0.22 | 0.09 |

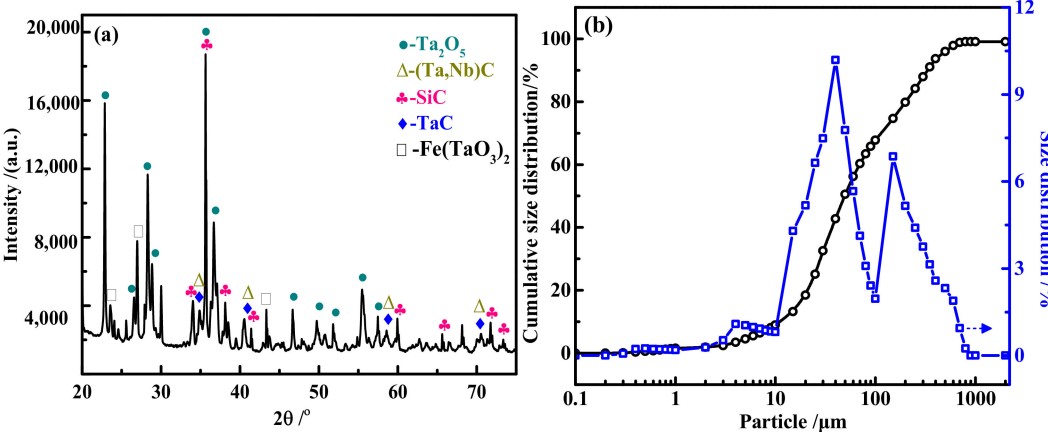

**Figure 1.** XRD pattern (**a**) and particle size distribution diagram (**b**) of the tantalum slag raw materials.

The oxygen, sodium hydroxide, hydrofluoric acid, potassium chloride, and octanol [32] used in the experiment were all analytically pure, and pure water was used as a stripping agent. The equipment used in the alkaline decomposition of Ta slag was an autoclave (500 mL, YZPR-500, Yanzhen Experimental Instrument Co., Ltd., Shanghai, China) with an inner lining of a tetrafluoroethylene cylinder, and the stirring impeller was made of stainless steel. The low-acid leaching and crystallization experiment was carried out in a magnetic thermostatic water bath (DF-101S, Lichen Instrument Technology,Changsha, China), which was held in a plastic beaker and stirred by magnetic particles. Extraction and stripping were carried out in a separation funnel with the extraction agent pure sec-octanol.

Inductively coupled plasma spectroscopy (ICP, Nexion 2000, PerkinElmer, Waltham, Massachusetts, USA) was used for the content analysis of low concentrations of Ta and Nb in solution. The phase composed of Ta slag and leaching residue was analyzed by X-ray diffraction (XRD) using a D/MAX 2500VB2+/PC diffractometer (Rigaku, Japan); the samples were scanned in steps of 0.02° in a 2θ range of 10–80. The morphology was

detected using a JSM-6360LV scanning electron microscope (SEM) instrument coupled with an EDX microanalyzer (JEOL, Tokyo, Japan).

### 2.2. Procedure and Methods

Tantalum-bearing waste slag was used as a raw material. The whole process included pressurized alkaline decomposition, low-acid leaching, solvent extraction, and crystallization. During the process, tantalum and impurities can be separated, and high-purity potassium fluotantalate products can be prepared, while low-content niobium can also be recovered. A suggested process route is shown in Figure 2. First, the tantalum slag was converted into sodium tantalate and sodium niobate by pressurized alkaline decomposition, which can be decomposed by low-concentration HF to obtain fluorotantalate and fluoroniobate. Then, because the content of tantalum in the tantalum waste slag washigher than that of niobium, the extraction of tantalum was first performed with low acid and was then back-extracted by pure water. Finally, potassium chloride was added to the stripping liquor to prepare potassium fluotantalate, and the product was obtained by evaporation crystallization. The niobium-containing residual raffinate was returned to the low-acid leaching process, and the cycle was repeated several times. After the niobium reached the appropriate concentration, the niobium hydroxide product could be obtained by acid extraction and pure water reverse extraction. The main reaction equations and detailed experiment methods are shown as follows.

$$Ta_2O_5 + 2NaOH = 2NaTaO_3 + H_2O \tag{1}$$

$$Ta_2O_5 + 6NaOH = 2Na_3TaO_4 + 3H_2O \tag{2}$$

$$Fe(TaO_3)_2 + 2NaOH = 2NaTaO_3 + FeO + H_2O \tag{3}$$

$$Fe(TaO_3)_2 + 6NaOH = 2Na_3TaO_4 + FeO + 3H_2O \tag{4}$$

$$NaTaO_3 + 6HF = NaTaF_6 + 3H_2O \tag{5}$$

$$Na_3TaO_4 + 8HF = NaTaF_6 + 2NaF + 4H_2O \tag{6}$$

(1) Decomposition: The pressure alkali decomposition of Ta slag was conducted in an autoclave, and a certain amount of sodium hydroxide solution was prepared after dissolution and cooling. A total of 50 g of tantalum slag and weighing NaOH solution were charged in the lining tube of the autoclave, heated to the setting temperature, stirred at 800 rpm, and oxygen was added at the pressure of the target value. After the reaction, cooling water was supplied to cool the reactor before removal. The reaction slurry was filtered, washed, and dried, and then alkali decomposition residue was obtained and weighed. The alkali decomposition residue was dissolved with hydrofluoric acid, 10 g of decomposed residue was put into a plastic beaker, and 50 mL of hydrofluoric acid (10 mol/L) was added and heated to 60 °C for 1 h. After filtration, the tantalum and niobium contents in the filtrate were analyzed by chromatography [33] and ICP-OES, and the tantalum and niobium decomposition efficiencies were calculated using the formula:

$$\text{Decomposition efficiency} = \frac{m_0 x_0 - \frac{m_1}{10} v_1 x_1}{m_0 x_0} \times 100 \tag{7}$$

where $m_0$ is the mass of Ta waste slag, g; $x_0$ is the content of Ta and Nb in Ta slag, %; $m_1$ is the mass of alkaline decomposition slag, g; $v_1$ is the volume of acid leaching solution, mL; and $x_1$ is the Ta content in acid leaching solution, g/L.

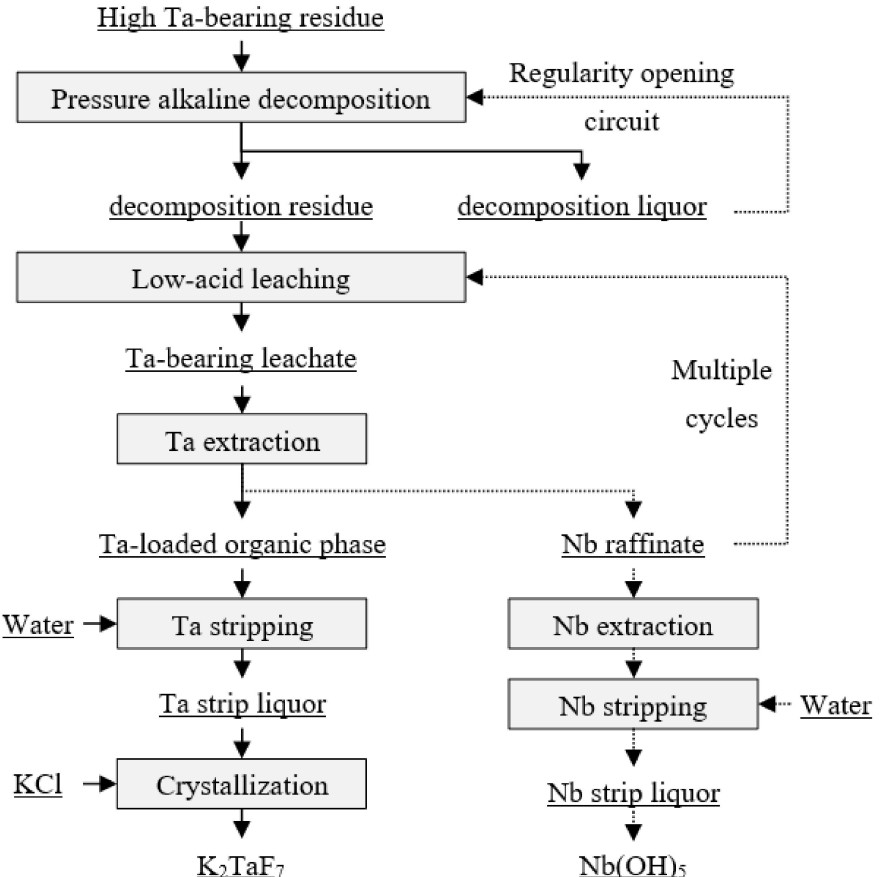

**Figure 2.** The process for recovery of tantalum slag by pressurized alkali leaching.

(2) Low-acid leaching: The low-acid leaching of alkaline decomposition residue was carried out in a water bath, and 50 g decomposition residue was added to a polytetrafluoron beaker. Then, different concentrations and amounts of hydrofluoric acid were added to investigate the leaching of tantalum and niobium. Similar formulas were used to calculate the leaching efficiencies of tantalum and Nb.

(3) Solvent extraction of Ta: The Ta extracted from the leaching solution was conducted in a separation funnel with a 100 mL solution every time. All tests were carried out at room temperature (~25 °C), and the time of oscillation and clarification was 1 min. Extraction conditions: pure sec-octanol reagent, O/A (volume ratio of the organic phase to the water phase) = 1.5:1; washing conditions: 1.25 mol/L sulfuric acid, O/A= 1:0.25; extraction conditions: pure water, O/A= 1:0.25. The separation factor was introduced to explain the separation result of Ta and Nb, and it was calculated using the formula:

$$\text{Separation factore} = \frac{x_2/x_1}{y_2/y_1} \tag{8}$$

where $x_1$ and $y_1$ are the Ta and Nb content in acid leaching solution, *g/L*. $x_2$ and $y_2$ are the Ta and Nb content in stripping solution, *g/L*.

(4) Crystallization: Potassium fluotantalate was prepared by the crystallization of potassium chloride; the Ta stripping liquor was heated to 90 °C, and the additional amounts of hydrofluoric acid and potassium chloride were 10 and 0.8 mole times Ta, respectively. Then, the potassium tantalum fluoride crystal was separated after cooling crystallization. The potassium fluotantalate crystal was washed with water and dried to obtain the finished product [34].

## 3. Results and Discussion

### *3.1. Pressure Alkaline Decomposition of Ta Slag*

#### 3.1.1. Effect of Time on Decomposition

The fixed reaction temperature was 200 °C, the liquid–solid ratio was 7:1, the oxygen partial pressure was 2 MPa, and the basicity (mass concentration of NaOH in solution, wt.%) was 40 wt.%. The effect of decomposition time on the decomposition efficiencies of tantalum and niobium was investigated, and the results are shown in Figure 3. The effect of decomposition time on the decomposition efficiencies of tantalum and Nbwas not obvious. The decomposition efficiencies of Ta and Nb both increased slowly over time before 2 h. After 2 h, the decomposition efficiencies of Ta and Nb remained constant, approximately 91.55% and 94.21%, respectively. In addition, the decomposition efficiency of Nbwas higher than that of Ta, indicating that Nbwas easier to decompose than Ta. Therefore, 2 h was considered the optimal decomposition time.

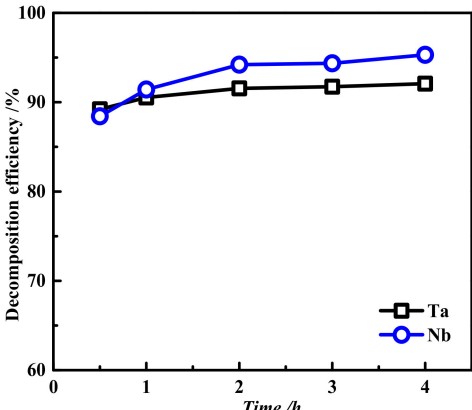

**Figure 3.** Effects of different decomposition times on the decomposition efficiencies of tantalum and Nb: reaction temperature 200 °C, liquid–solid ratio 7:1, oxygen partial pressure 2 MPa, and basicity 40 wt.%.

The alkaline decomposition residue is the intermediate for tantalum and Nb recovery, so the decomposition degree of Ta slag will directly determine the recovery rate of Ta and Nb. Phase analysis was carried out on the alkaline decomposition residue, and the results are shown in Figure 4. This shows that the main composition of the decomposition slag was sodium metatantalate, with a small amount of tantalum pentoxide. Therefore, through alkaline pressure decomposition, the tantalum carbide, ferric tantalate, and tantalum oxide in the raw material were transformed into sodium metatantalate, indicating that the decomposition reaction was sufficient.

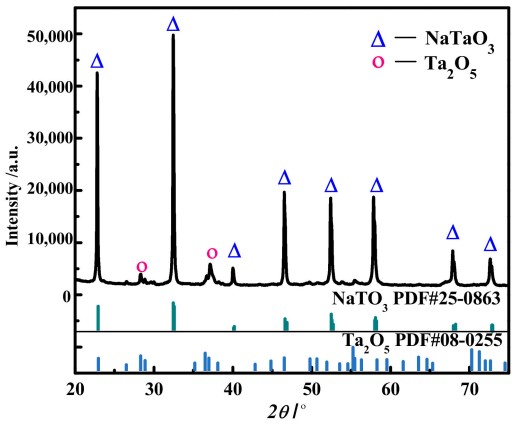

**Figure 4.** XRD diagram of the decomposed slag.

3.1.2. Effect of Oxygen Particle Pressure on Decomposition

The fixed reaction time was 2 h, the reaction temperature was 200 °C, the liquid–solid ratio was 7:1, the basicity was 40 wt.%, and the effect of oxygen partial pressure on Ta and Nb decomposition was investigated in the range of 1.5 MPa to 4.0 MPa, the results are shown in Figure 5. This shows that with the increase in oxygen partial pressure in the autoclave, the tantalum niobium decomposition efficiency increased, but the increment was small. When the oxygen partial pressure was 2.5 MPa, the decomposition efficiencies of tantalum and Nb were almost stable, remaining at approximately 91% and 93%, respectively. The increase in oxygen partial pressure canforce the decomposition reaction of tantalum and Nb to the right and promote the decomposition of Ta and Nb, thus improving the decomposition efficiencies of Ta and Nb. As the oxygen partial pressure continued to increase, the decomposition reaction of tantalum and Nb gradually reached equilibrium, so the effect of oxygen partial pressure on the decomposition efficiencies of Ta and Nbwas gradually weakened. Overall, the partial pressure of oxygen was 2.5 MPa, which met the oxygen pressure required for alkaline pressure decomposition.

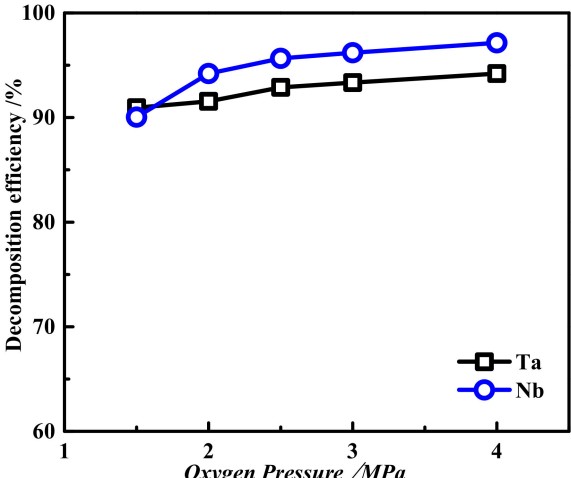

**Figure 5.** The effect of oxygen partial pressure on the decomposition efficiencies of tantalum niobium: reaction time 2 h, reaction temperature 200 °C, liquid–solid ratio 7:1, and basicity 40 wt.%.

SEM and Energy Dispersive X-Ray Spectroscopy(EDX) analyses were performed on the decomposed slag obtained under different oxygen partial pressures, and the results are shown in Figure 6 and Table 2. At an oxygen partial pressure of 1.5 MPa, there were obvious lumpy grains in the decomposed slag, with a compact surface and regular structure. The content of niobium was not detected at high oxygen partial pressures of 2.5MPa by EDX analysis, which indicates that niobium is easy to decompose, and the decomposed slag also contains organic matter, Zr, and other components. However, in the decomposition slag of 2.5 MPa, there were fewer large particles with compact surfaces than that of 1.5 MPa. Compared with 2.5 MPa, the particles of decomposed slag at 3 MPa and 4 MPa were finer and mainly composed of loose particles, while some coarse particles of tantalum particles were still not decomposed, mainly in the oxidation state. Therefore, with the increase in oxygen partial pressure, the particle size of decomposed slag becomes finer, and the disappearance of massive particles indicates that the decomposition reaction is more sufficient.

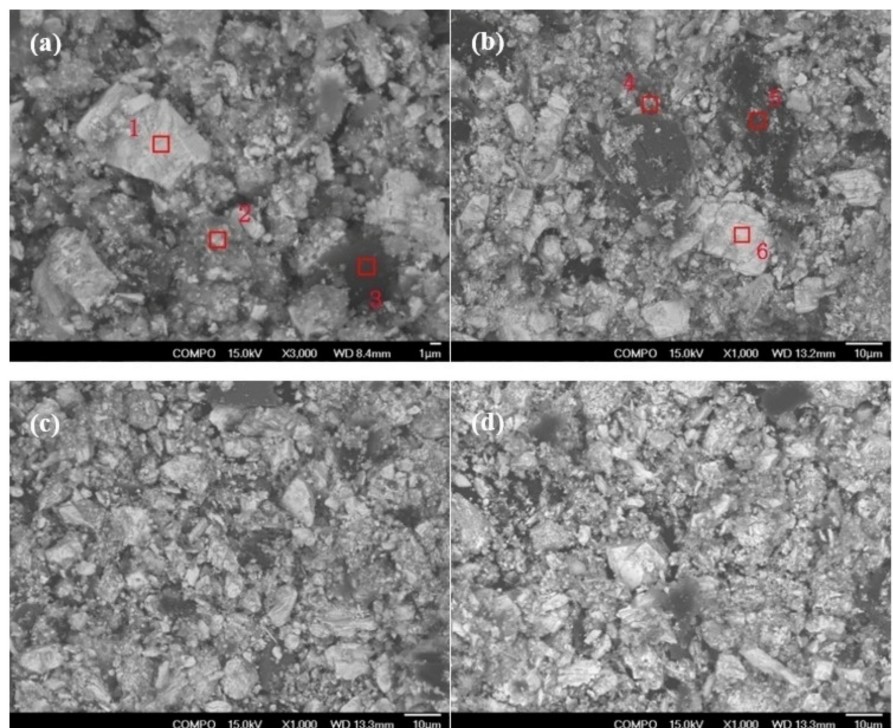

**Figure 6.** SEM diagram of decomposed slag under different oxygen partial pressures. (**a**) 1.5 MPa, (**b**) 2.5 MPa, (**c**) 3 MPa, and (**d**) 4 MPa.

**Table 2.** EDX results of decomposed slag under different oxygen partial pressures.

| No. | Ta | Nb | Zr | Ca | Na | F | O | C | Ti | Al |
|-----|-------|------|------|------|------|------|-------|-------|------|-------|
| 1 | 74.08 | 0.18 | - | 1.45 | 0.48 | 4.12 | 15.1 | 4.59 | - | - |
| 2 | 55.74 | 2.68 | 3.24 | 5.01 | 1.1 | 9 | 12.38 | 10.85 | - | - |
| 3 | 24.01 | 1.17 | 1 | 2.21 | 0.5 | 4.54 | 7.19 | 59.39 | - | - |
| 4 | 8.88 | - | - | 0.48 | - | 1.5 | 30.09 | 10.12 | 0.86 | 48.07 |
| 5 | 12.69 | - | - | 1.1 | - | 3.48 | 5.33 | 77.4 | - | - |
| 6 | 76.08 | - | - | 2.29 | - | 5.72 | 11.52 | 4.4 | - | - |

### 3.1.3. Effect of the Liquid–Solid Ratio on Decomposition

With a fixed decomposition time of 2 h, oxygen partial pressure of 2.5 MPa, basicity of 40 wt.%, and a temperature of 200 °C, the influence of the liquid–solid ratio on the tantalum and niobium decomposition efficiencies was investigated, and the results are shown in Figure 7. As shown in Figure 7, the liquid–solid ratio had little influence on the decomposition efficiencies of tantalum and Nb. With the increase in the liquid–solid ratio, the decomposition efficiencies of tantalum and Nb remained unchanged, both remaining above 90%. A large liquid–solid ratio will reduce the reaction efficiency, so a liquid–solid ratio of 4:1 was selected for the next test.

### 3.1.4. Effect of Basicity on Decomposition

The fixed reaction temperature was 200 °C, the liquid–solid ratio was 4:1, the oxygen partial pressure was 2.5 MPa, the time was 2 h, and the NaOH concentration was 20–60 wt.%. The influence of basicity on the decomposition efficiencies of tantalum and niobium was investigated, and the results are shown in Figure 8a. As shown in Figure 8, with increasing basicity, the decomposition efficiencies of tantalum and niobium first increased and then decreased. When the basicity was 40 wt.%, the decomposition efficiencies of tantalum and Nb reached maximum values of 93.62% and 95.42%, respectively. When the basicity exceeded 40 wt.%, both decompositions began to decrease. However, the basicity

had a great influence on the niobium decomposition efficiency. Before 40 wt.% basicity, with increasing basicity, the niobium decomposition efficiency increased rapidly from 25.7% to 95.42% and then decreased rapidly to 29.15% at 60 wt.% basicity. Meanwhile, the content of Nb and Ta increased from 0.33 g/L to 6.17 g/L and 13.49 g/L to 28.16 g/L when the basicity was increased from 40 wt.% to 60 wt.%. Therefore, the dissolved amount of tantalum and niobium increased with the increase of NaOH. This is because when the basicity was too low, the conversion amount of tantalum and niobium to sodium metatantalate and sodium metaniobatewas less. Therefore, 40 wt. % basicity is the best choice.

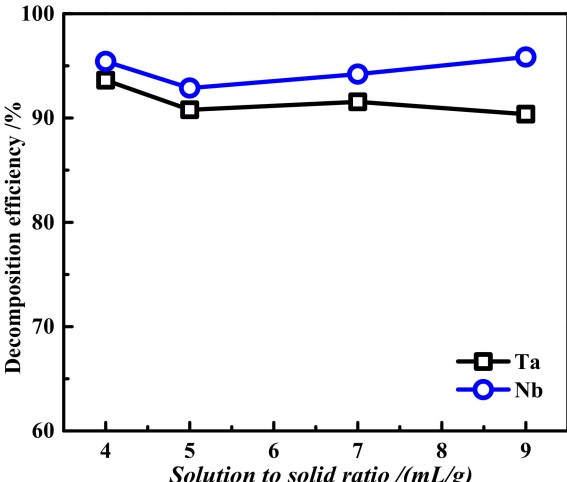

**Figure 7.** Effect of different liquid–solid ratios on the decomposition efficiencies of tantalum and niobium: decomposition time 2 h, oxygen partial pressure 2.5 MPa, basicity 40 wt.%, and reaction temperature 200 °C.

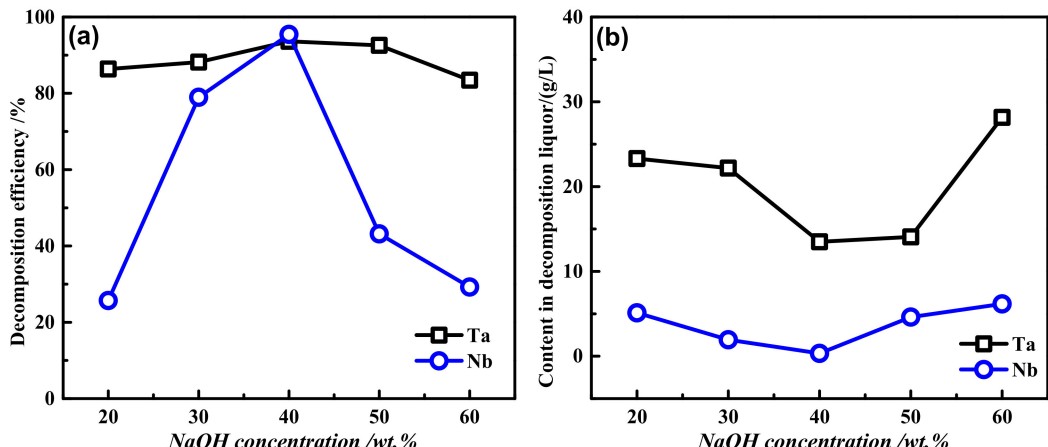

**Figure 8.** The effect of different alkalinities on the decomposition efficiencies (**a**) and dissolution (**b**) of tantalum niobium: reaction temperature 200 °C, liquid–solid ratio 4:1, oxygen partial pressure 2.5 MPa, and reaction time 2h.

### 3.1.5. Effect of Temperature on Decomposition

Fixed basicity 40 wt.%, liquid–solid ratio 4:1, oxygen partial pressure 2.5 MPa for 2 h, and the effect of temperature on the decomposition efficiency of tantalum and Nbwereinvestigated. As shown in Figure 9, with increasing temperature, the decomposition efficiencies of tantalum and Nb first increased and then decreased, reaching their maximum values at 200 °C. As the temperature continued to rise, the tantalum and niobium decomposition efficiencies began to decline. This is because, at low temperatures, the

reaction rate between the material and lye is slow, making the tantalum and niobium decomposition incomplete, resulting in low decomposition efficiencies of tantalum and niobium. However, a high reaction temperature will accelerate the reaction rate but cause the solubility of tantalum and niobate in the solution, thus reducing the amount of slag and affecting the decomposition efficiencies during the transformation [34]. Therefore, the optimal decomposition temperature was 200 °C.

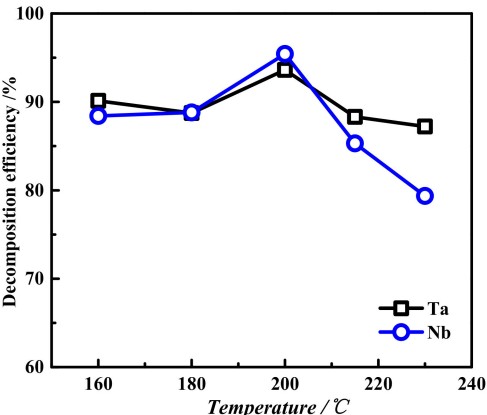

**Figure 9.** The effect of different temperatures on the decomposition efficiencies of tantalum niobium: basicity 40 wt.%, liquid–solid ratio 4:1, oxygen partial pressure 2.5 MPa, and reaction time 2 h.

### 3.2. Low-Acid Leaching of Alkali Decomposition Residue

The tantalum and niobium in the slag can be changed into sodium tantalate and sodium niobate by pressure alkaline decomposition. Then, it can be transformed into soluble fluoroniobate and fluorotantalate with low concentrations of HF, which can greatly reduce the consumption of HF compared with the current industrial practice (approximately 24 mol/L). To accurately study the influence of acidity (the mole concentration of HF in solution, mol/L) and other conditions on the leaching of sodium tantalate and sodium niobate, pure sodium tantalate and sodium niobate prepared by pure reagent were first used for HF leaching to eliminate the influence of impurities. The XRD and SEM of the synthesized sodium tantalate and sodium niobate are shown in Figure 10.

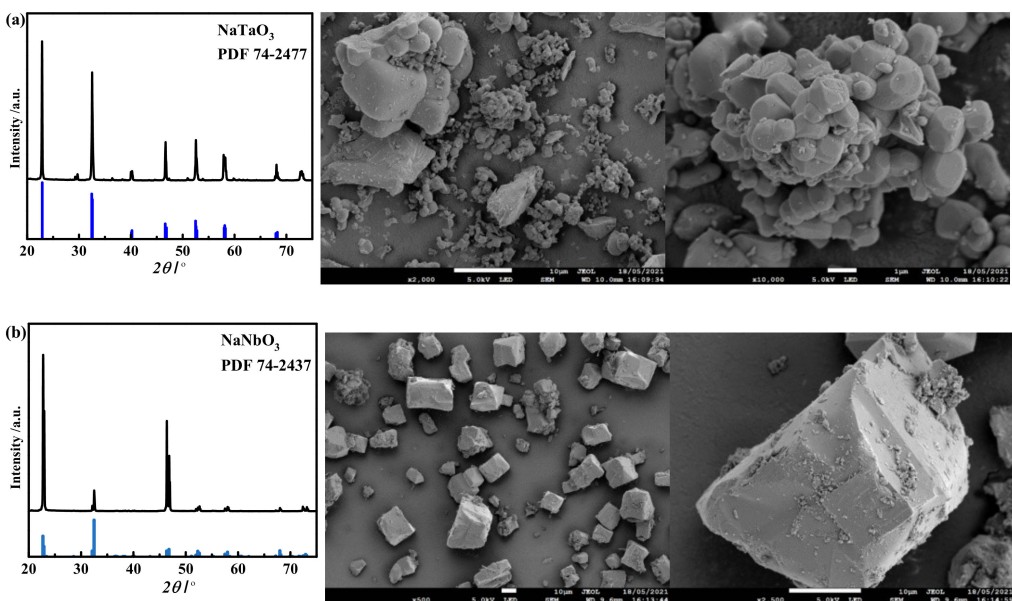

**Figure 10.** XRD and SEM images of synthesized sodium tantalate (**a**) and sodium niobate (**b**).

As seen from the figure, pure $NaTaO_3$ and $NaNbO_3$ were prepared at $nTa_2O_5/nNa_2CO_3$ =1:1 and $nNb_2O_5/nNa_2CO_3$=1:1, and the XRD peaks of the samples matched perfectly with the standard cards. From the SEM figure, it can be seen that the synthesized sodium tantalatewas composed of spheroid particles, and the particles were clumped together. Sodium niobatewas mainly composed of larger square particles with relatively complete crystal shapes and less agglomeration and accumulation between particles, which was not conducive to the leaching of sodium niobate. Therefore, sodium tantalate may decompose more easily than sodium niobate. The synthesized sodium tantalate and sodium niobate were subjected to HF leaching, and the results are shown in Figure 11. It can be seen from the figure that the HF concentration, reaction temperature, and time all have a great influence on the leaching efficiencies of Ta and Nb. Figure 11a show that the HF concentration had little influence on tantalum, and the leaching efficiency of tantalum reached 97.37% at an HF concentration of 2 mol/L, while that of niobium was only 54.29%. With increasing HF acid concentration, the leaching efficiency of niobium increased rapidly, reaching 99.06% at 6 mol/L HF. The leaching results indicate that tantalum is easier to leach than Nb, which is consistent with the morphology analysis results in Figure 10. It can be seen that the high leaching efficiencies of sodium tantalate and sodium niobate can be achieved at a concentration far lower than the current industrial practice.

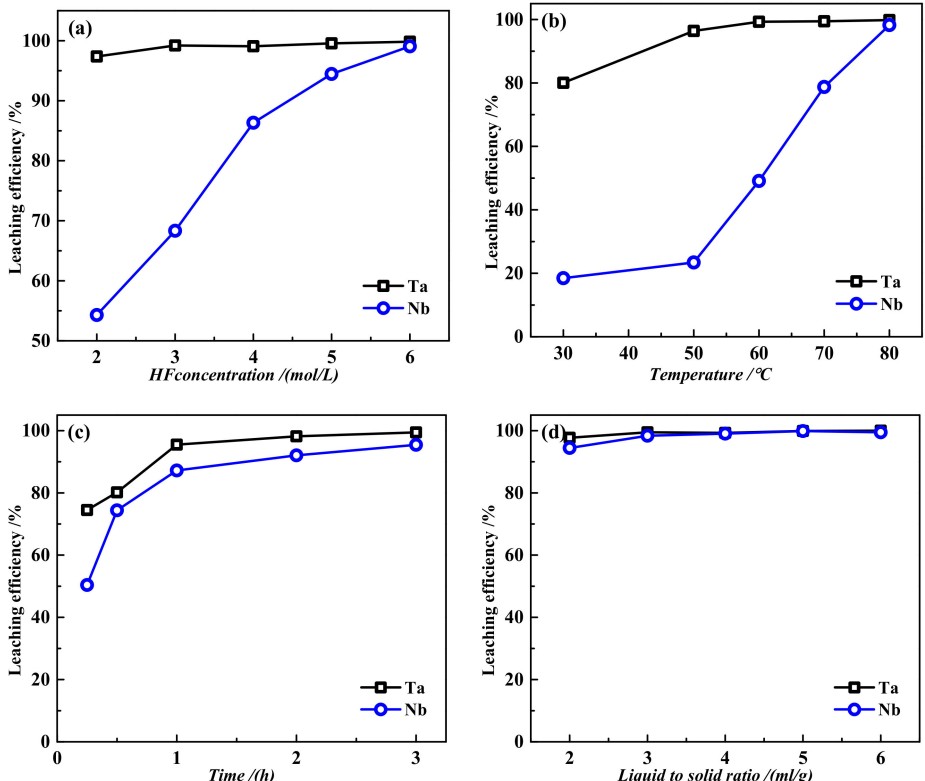

**Figure 11.** Effect of the different conditions on leaching of alkaline decomposition residue: (**a**) temperature 60°C, time 1h, liquid–solid ratio 4mL/g; (**b**) HF 5 mol/L, time 1h, liquid–solid ratio 4mL/g; (**c**) HF 5 mol/L, temperature 70 °C, liquid–solid ratio 4mL/g; (**d**) HF 5 mol/L, temperature 70°C, time 1h.

Figure 11b show that temperature greatly influences the leaching of tantalum and niobium. At low temperatures, the leaching efficiencies of tantalum and Nb are not high. When the temperature is above 50 °C, the leaching efficiency of tantalum increases sharply and gradually becomes stable, while the leaching efficiency of niobium continues to rise sharply. Therefore, the temperature increase is beneficial to the leaching of tantalum and Nb, and the leaching efficiencies of both Ta and Nb are over 99% at 80 °C. From Figure 11c,

it can be seen that the leaching time has a consistent effect on the Ta and Nb. With time extension, the leaching efficiencies increase and gradually become stable. Finally, the liquid–solid ratio has little effect on the leaching rates of sodium tantalate and sodium niobate. Even at a liquid–solid ratio of approximately 3:1, the leaching rate of tantalum and niobate can be over 98%. Therefore, the suitable low-acid leaching conditions are HF 2 mol/L for sodium tantalate and 6 mol/L for sodium niobate, a temperature of 80 °C, a time of 2 h, and a liquid-to-solid ratio of 4:1. The difference in leaching acidity between sodium tantalate and sodium niobate is also beneficial for the selective leaching separation of tantalum and niobate.

Therefore, the optimal conditions for NaOH decomposition of Ta slag and HF leaching of decomposition residue were obtained from the above two sections. The whole process comprehensive test was performed with a scale of 500 g tantalum slag, and the results of the element balance analysis are shown in Table 3. Firstly, as seen from the table, the decomposition efficiencies of tantalum and niobium in the comprehensive test still reached 93.13% and 95.27%, respectively, and more than 60% of silicon entered the decomposition liquor to achieve the effect of desilication. The decomposition liquor contained 9.85 g/L tantalum, and the other main components were excess alkali. Therefore, the decomposition liquor returned to the decomposition process of alkali pressure leaching. Secondly, in the process of low-acid leaching the decomposition residue, the leaching efficiencies of tantalum and niobium were 98.14% and 96.26%, respectively, and the concentrations in the leaching solution were 123.43 g/L and 5.85 g/L, respectively. According to the calculation, from the tantalum slag to the low-acid leaching solution, the comprehensive leaching ratios of Ta and Nbwere 91.40% and 91.71%, respectively. Therefore, the low-acid consumption and high-efficiency leaching of Ta and Nb can be realized, and the residual Ta and Nb can be recovered by a recycling process. The concentration of Ta in the low-acid leaching solution was much higher than that of other impurity ions, including Nb, so it was necessary to prioritize Ta extraction.

### 3.3. Ta Extraction from the Low-Acid Leaching Solution

The low-acid leaching solution was subjected to Ta extraction and $K_2TaF_7$ preparation, using sec-octanol as the extraction agent and HF and $H_2SO_4$ as the acidity regulator. The whole solvent extraction process, including one extraction and one reverse extraction, and the extraction and separation of tantalum under different acidities were investigated. The results are shown in Table 4. Ta, Nb, and other impurities were separated thoroughly. When the HF acidity was 1 mol/L, the Ta content in the reflux solution reached 77.38 g/L, and the niobium content was only 0.024 g/L. The separation factor of tantalum and Nb reached over 200. However, with the increasing HF concentration, the extraction rate of tantalum decreased, the niobium extraction rate increased, and the silicon content in the reverse extraction liquid increased. Therefore, low acidity is favorable for tantalum extraction and obtaining pure tantalum liquid.

**Table 3.** Results of the comprehensive experiment of low-acid leaching of decomposition residue.

| Element | Input | Decomposition Liquor (1817 mL) | | | Decomposition Residue (629 g) | | | Low-Acid Leaching Liquor (2681 mL) | | | Low-Acid Leaching Residue (87.15 g) | | |
|---|---|---|---|---|---|---|---|---|---|---|---|---|---|
| | Weight (g) | Weight (g) | Concentration (g/L) | Distribution ratio (%) | Weight (g) | Content (%) | Distribution ratio (%) | Weight (g) | Concentration (g/L) | Distribution ratio (%) | Weight (g) | Concentration (%) | Distribution ratio (%) |
| Ta | 362.05 | 17.89 | 9.85 | 4.94 | 337.18 | 53.61 | 93.13 | 330.91 | 123.43 | 98.14 | 3.14 | 3.60 | 0.93 |
| Nb | 17.1 | 0.93 | 0.51 | 5.44 | 16.29 | 2.59 | 95.27 | 15.68 | 5.85 | 96.26 | 0.29 | 0.33 | 1.78 |
| Ca | 13.05 | 0.093 | 0.051 | 0.71 | 12.95 | 2.05 | 99.27 | 11.49 | 4.29 | 88.77 | 1.55 | 1.78 | 11.96 |
| Ti | 9.4 | 1.79 | 0.99 | 19.04 | 8.18 | 1.30 | 87.07 | 5.49 | 2.04 | 67.11 | 2.96 | 3.39 | 36.16 |
| Si | 7.2 | 4.33 | 2.38 | 60.14 | 3.25 | 0.52 | 45.23 | 3.00 | 1.11 | 92.13 | 0.44 | 0.50 | 13.51 |
| Fe | 6.35 | 0.27 | 0.15 | 4.25 | 6.18 | 0.98 | 97.35 | 2.28 | 0.85 | 36.92 | 4.04 | 4.63 | 65.35 |
| Cu | 1.1 | 0.075 | 0.041 | 6.82 | 1.04 | 0.16 | 94.8 | 0.49 | 0.18 | 47.24 | 0.6 | 0.68 | 57.54 |
| Al | 0.45 | 0.42 | 0.23 | 93.33 | 0.065 | 0.010 | 14.63 | 0.044 | 0.016 | 66.25 | 0.02 | 0.023 | 30.38 |

**Table 4.** Ta extraction from the low-acid leaching solution.

| HF Concentration (mol/L) | Operating Parameters | Stripping Solution (g/L) | | | | | |
|---|---|---|---|---|---|---|---|
| | | Ta | Nb | Fe | Ti | Ca | Si |
| 1 | Extraction: octanol, phase ratio O/A=1.5:1; Washing: 1.25 moL/L $H_2SO_4$, phase ratio O/A=1:0.25; Stripping: pure water, phase ratio O/A=1:1 | 77.38 | 0.024 | 0.0003 | 0.0004 | 0.0004 | 0.003 |
| 2 | | 76.89 | 0.11 | 0.0009 | 0.0005 | 0.0008 | 0.009 |
| 3 | | 76.45 | 0.21 | 0.0005 | 0.0008 | 0.0007 | 0.013 |

The acidity of the stripping solution was adjusted to 1.8 mol/L by hydrofluoric acid, the solution was heated above 95 °C, and the reagent potassium chloride was added at 0.85 times the $Ta_2O_5$ mass to precipitate the tantalum. After crystallization, cooling, washing, and drying, the potassium fluorotantalate samples were prepared. The purity analysis results are shown in Table 5. The content of all impurity elements was below 10 ppm, and the purity of $K_2TaF_7$ was greater than 99.99%, which met the requirements of commercial-grade purity. Figure 12 show the XRD pattern and SEM image of the $K_2TaF_7$ product. It can be observed that all of the peaks of the sample were identified as $K_2TaF_7$ (JCPDS card, No.19-0997), indicating that the composition of the sample was pure. The images of $K_2TaF_7$ were also studied. The images show that the particles are mainly square and have a large size.

**Table 5.** Purity of the potassium fluorotantalate product (unit: ppm).

| Element | Ca | Cu | Cr | Fe | Mg | Mo | Nb | Ni | Pb | Si |
|---|---|---|---|---|---|---|---|---|---|---|
| Content | - | <0.5 | <4 | <4 | <1 | <4 | <4 | <4 | <4 | <5 |
| Element | Ti | W | Zr | Ca | Al | V | Sn | Zn | $K_2TaF_7$ | |
| Content | <3 | <0.5 | <0.5 | <2 | <0.5 | <0.5 | <0.5 | <0.5 | >99.99% | |

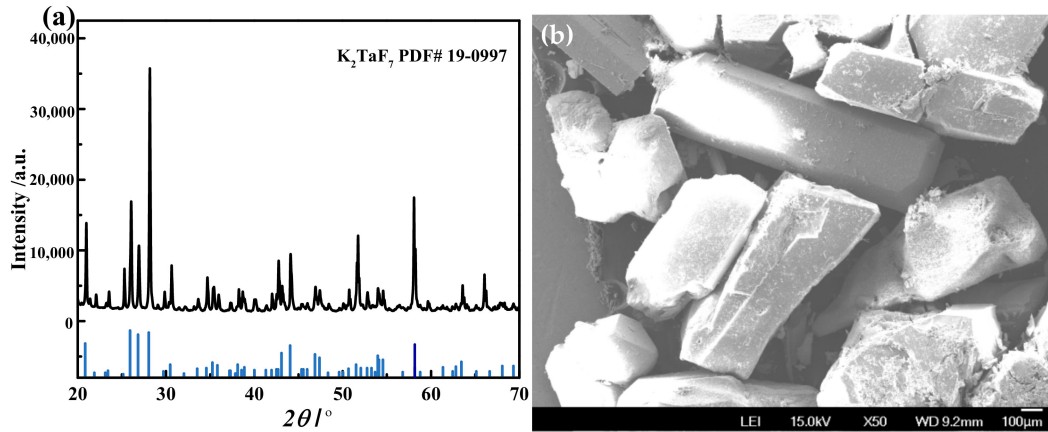

**Figure 12.** The XRD pattern (**a**) and SEM image (**b**) of thepotassiumfluorotantalate product.

## 4. Conclusions

The whole process, including alkali pressure decomposition, low-acid leaching, and tantalum extraction and crystallization, was proposed for the recovery of tantalum from tantalum slag and the preparation of potassium fluotantalate. First, tantalum slag was subjected to alkali pressure decomposition under the best conditions of a reaction time of 2 h, oxygen partial pressure of 2.5 MPa, liquid–solid ratio 4:1, basicity 40 wt.%, and decomposition temperature 200 °C, where the decomposition efficiencies of tantalum and Nb reached 93.62% and 95.42%, respectively. With the increase in oxygen partial pressure,

the particle size of the decomposed slag was finer, and the decomposition was more complete. The main component of the decomposed slag was sodium tantalate. Second, the optimal conditions for low-acid leaching of sodium tantalum niobatewere obtained as follows: HF acid 3 mol/L ~ 6 mol/L, temperature 80 °C, time 2 h, liquid–solid ratio 4:1. The acidity was selected according to the content of tantalum and Nb, and the leaching efficiencies of Ta and Nbwere more than 99%. Last, under the low acid conditions of 1 mol/L HF and 1 mol/L $H_2SO_4$, the tantalum extraction rate and tantalum and niobium separation factors were above 94% and 200, respectively. The whole process realized high purity potassium fluotantalate preparation from tantalum slag. Compared with current industrial practice, the consumption of hydrofluoric acid was greatly reduced, and the recovery rate of tantalum was increased.

**Author Contributions:** K.X. and M.W. carried out the experimental research and collected the data, L.Y. designed the research and wrote the paper, and X.W., J.W. and S.L. reviewed and contributed to the final manuscript. All authors have read and agreed to the published version of the manuscript.

**Funding:** This research was funded by The National Key Research and Development Program of China(Grant No. 2018YFC1901701).

**Data Availability Statement:** Not applicable.

**Acknowledgments:** This project was supported financially by The National Key Research and Development Program of China(Grant No. 2018YFC1901701), for which the authors are grateful. We also acknowledge the several helpful comments and suggestions from anonymous reviewers.

**Conflicts of Interest:** The authors declare no conflict of interest.

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
