# Peer review of "Recovery and Preparation of Potassium Fluorotantalate from High-Tantalum-Bearing Waste Slag by Pressure Alkaline Decomposition"

_metals, doi:10.3390/met12040648_

Round 1

Reviewer 1 Report

This paper describes a Ta and Nb recycling process and the processing parameters were optimized based on their experimental results. This paper is worthy of publication in Metals after minor revisions.

(1)Clarify and describe the significant figure and describe the reproducibility of evaluated values, for instance, the leaching efficiencies such as 97.37%, 99.06%.

(2)Clarify the definition and the meaning of some words, e.g., the basicity, the alkalinity, the acidity. Did the authors use these words instead of solution concentration?

(3)There are some typographic errors. There are two "Figure 10". At Fig. 6, the positions of (a), (b), (c), (d) in this manuscript are not appropriated.

(4)What does the separation coefficient mean?

(5)Why Table 3 is placed after Table 4in this manuscript?

Author Response

Reviewer #1: This paper describes a Ta and Nb recycling process and the processing parameters were optimized based on their experimental results. This paper is worthy of publication in Metals after minor revisions.

(1) Clarify and describe the significant figure and describe the reproducibility of evaluated values, for instance, the leaching efficiencies such as 97.37%, 99.06%.

[Answer]: All the figures are reasonable and valid, and the Re-manuscript also unified the significant figures of the main results.

(2) Clarify the definition and the meaning of some words, e.g., the basicity, the alkalinity, the acidity. Did the authors use these words instead of solution concentration?

[Answer]: Both basicity and alkalinity are the meanings of the concentration of NaOH, and the acidity means the concentration of HF and H2SO4 in solution. The definition for them is added in the text.

(3) There are some typographic errors. There are two "Figure 10". At Fig. 6, the positions of (a), (b), (c), (d) in this manuscript are not appropriated.

[Answer]: They were revised.

(4) What does the separation coefficient mean?

[Answer]: It has been modified as a separation factor, and its means the ratio of distribution ratio tantalum to niobium. Its definition is given in the experimental section.

(5) Why Table 3 is placed after Table 4 in this manuscript?

[Answer]: This is mainly to consider the convenience of paper typesetting, and now it has been adjusted.

Reviewer 2 Report

Comments: 

Page 2, line 93-94: This needs more description.

Page 3, Excess lye? Authors are suggested to clarify the meaning of lye? Slag or slurry?

Page 3, line 103: “cancel the enrichment process”, what is enrichment process?

Page 3, line 112, what is the significance of hydrogenation pulverization?

Section 2.2 and 2.3 can be combined into one section.

Page 4, line 156; …filtrate were analyzed by chromatography, however, there is no description of this method in the whole manuscript.

Page 5, Please use organic phase or extractant phase instead of oil phase.

Page 5: Solvent extraction and crystallization: What was the basis for the selection of operating conditions used for these experiments? Reference should be provided.

Line 174: O/A (volume ratio of the water phase to the oil phase) = 1.5:1. It should be volume ratio of the oil phase to the water phase. Please correct.

Page 5, line 183-185: Should be moved to materials and instruments section

Authors are suggested to provide the complete experimental conditions in each figure's caption.

Page 6, line 207: what were the experimental conditions for the XRD diagram shown in Fig.4?

The analyses error, uncertainty or error bar should be added.

Line 235-236..content of niobium is much lower than that of tantalum by EDX analysis, which also indicates that niobium is easier to decompose. Since Nb conc in slag is low, it is obvious that Nb conc in residue would be low. Please Justify how EDX can indicate the easier decomposition.

Authors are suggested to carefully indicate the numbering in SEM figures.

Page 7, line 238-245: “However, in the decomposition slag of 2.5 MPa, there are fewer large particles with compact surfaces than that of the 1.5 MPa, while the EDX results show that the content of carbon in the particles is higher than that of 1.5 MPa, which hinders diffusion and affects the decomposition reaction.” – Why is carbon content higher in decomposition residue obtained at 2.5 MPa? Ideally, both should contain same carbon content as the feed material is same.

Page 8, line 276-278: XRD analysis of decomposition residue and the amount of metals dissolved in alkali solution should be included at different alkali concentration. This will provide evidence if the decrease in Ta and Nb dissolution at high alkali concentration is due to formation of insoluble phases or is dissolving in alkali solution or just the viscosity of slurry as mentioned in the text.

Page 9, line 292-295: Needs better explanation. It is mentioned that high temperature will cause solubility of Nb and Ta in the solution during alkali decomposition. Can you provide the exact values for metals dissolved in alkali solution at different temperatures to justify the claim?

Please rewrite the whole paragraph (line 351-369) and include all the experimental conditions properly for better understanding.

Table 3: In input material, what does two different weight fractions signify? Final product (Potassium fluorotantalate) can be included in the material balance.

Table 4: The final product is potassium fluorotantalate, while Table 4 mentions Ta2O5 and no amount of F. How was the composition converted to oxides? It will be worth adding XRD and SEM of the final product as well.

Potassium fluotantalate should be replaced with potassium fluorotantalate.

Mention the phase ratio used for the solvent extraction (Table 4).

Authors are suggested to include the SX plots instead of Table. It is puzzling to understand.

Author Response

(1) Page 2, line 93-94: This needs more description.

[Answer]: It was revised. In the tantalum and niobium profiles and purity process in the bombarding furnace, due to bombardment of electron beam and high temperature volatilization, a certain amount of metals are deposited on the inner wall of the furnace, so the hanging fireplace slag is produced

(2) Page 3, Excess lye? Authors are suggested to clarify the meaning of lye? Slag or slurry?

[Answer]: It was revised. The decomposed lixivium contained excess alkali unreacted can be returned to decomposition for recycling

(3) Page 3, line 103: “cancel the enrichment process”, what is enrichment process?

[Answer]: It was revised to " it can cancel the alkali enrichment process from decomposed lixivium and reduce alkali consumption "

(4) Page 3, line 112, what is the significance of hydrogenation pulverization?

[Answer]: Because the Ta slag is a hard lump, it can be pulverized by hydrogenation, which facilitates subsequent decomposition reactions.

(5) Section 2.2 and 2.3 can be combined into one section.

[Answer]: They were combined into one section 2.2.

(6) Page 4, line 156; …filtrate were analyzed by chromatography, however, there is no description of this method in the whole manuscript.

[Answer]: The chromatography is a conventional method for the analysis of tantalum and niobium, so it is not described in detail and references are added.

(7) Page 5, Please use organic phase or extractant phase instead of oil phase.

[Answer]: It was revised.

(8) Page 5: Solvent extraction and crystallization: What was the basis for the selection of operating conditions used for these experiments? Reference should be provided.

[Answer]: Ok , the reference was added.

(9) Line 174: O/A (volume ratio of the water phase to the oil phase) = 1.5:1. It should be volume ratio of the oil phase to the water phase. Please correct.

[Answer]: Thanks , it was revised.

(10) Page 5, line 183-185: Should be moved to materials and instruments section

[Answer]: Thanks , it was revised.

(11) Authors are suggested to provide the complete experimental conditions in each figure's caption.

[Answer]: Thanks , it was revised.

(12) Page 6, line 207: what were the experimental conditions for the XRD diagram shown in Fig.4?

The analyses error, uncertainty or error bar should be added.

[Answer]: The experimental conditions for the XRD diagram shown in Fig.4 is added in section 2.1. This result is carried out by XRD and error analysis cannot be carried out, but the standard cards of two phase were added

(13) Line 235-236..content of niobium is much lower than that of tantalum by EDX analysis, which also indicates that niobium is easier to decompose. Since Nb conc in slag is low, it is obvious that Nb conc in residue would be low. Please Justify how EDX can indicate the easier decomposition. Authors are suggested to carefully indicate the numbering in SEM figures.

[Answer]: It was revised. "The content of niobium is not detected at high oxygen partial pressures of 2.5MPa by EDX analysis, which indicates that niobium is easy to decompose."

(14) Page 7, line 238-245: “However, in the decomposition slag of 2.5 MPa, there are fewer large particles with compact surfaces than that of the 1.5 MPa, while the EDX results show that the content of carbon in the particles is higher than that of 1.5 MPa, which hinders diffusion and affects the decomposition reaction.” – Why is carbon content higher in decomposition residue obtained at 2.5 MPa? Ideally, both should contain same carbon content as the feed material is same.

[Answer]: The expression was deleted.

(15) Page 8, line 276-278: XRD analysis of decomposition residue and the amount of metals dissolved in alkali solution should be included at different alkali concentration. This will provide evidence if the decrease in Ta and Nb dissolution at high alkali concentration is due to formation of insoluble phases or is dissolving in alkali solution or just the viscosity of slurry as mentioned in the text.

[Answer]: With the increase of alkali concentration, the decomposition efficiency of tantalum does not decrease obviously, but the content of niobium decreases obviously. The XRD cannot detect this difference. The change in Ta and Nb content in solution has added.

(16) Page 9, line 292-295: Needs better explanation. It is mentioned that high temperature will cause solubility of Nb and Ta in the solution during alkali decomposition. Can you provide the exact values for metals dissolved in alkali solution at different temperatures to justify the claim?

Please rewrite the whole paragraph (line 351-369) and include all the experimental conditions properly for better understanding.

[Answer]: In general, the solubility of endothermic substances will increase with the temperature increasing. This point is confirmed by referring to the relevant data manual and also introduced in the references. Paragraph (line 351-369) was rewrote.

(17) Table 3: In input material, what does two different weight fractions signify? Final product (Potassium fluorotantalate) can be included in the material balance.

[Answer]: There is only one weight fraction, the another is the weight of element in decomposition liquor. Because the table doesn't have an inner box.

Your suggestion is very good, but from the leach solution to the potassium fluorotantalate product, there are multiple process included extraction, washing, backextraction, and  crystallization, these contents are too many to be presented in a table, and they are also not the main research content, so they do not appear in this part.

(18) Table 4: The final product is potassium fluorotantalate, while Table 4 mentions Ta2O5 and no amount of F. How was the composition converted to oxides? It will be worth adding XRD and SEM of the final product as well. Potassium fluorotantalate should be replaced with potassium fluorotantalate. Mention the phase ratio used for the solvent extraction (Table 4). Authors are suggested to include the SX plots instead of Table. It is puzzling to understand.

[Answer]: 1) The final product is not oxide, but the industry is used to indicate the purity of the potassium fluorotantalate product by the content of tantalum oxide, it has been changed to product purity in the form of potassium fluorotantalate; 2) The XRD and SEM of the final product were added; 3) It was revised; 4) The table was changed.